# Effects of Khat on Surface Roughness and Color of Feldspathic and Zirconia Porcelain Materials under Simulated Oral Cavity Conditions

**DOI:** 10.3390/medicina56050234

**Published:** 2020-05-13

**Authors:** Mohammed M. Al Moaleem, Rashad AlSanosy, Nasser M. Al Ahmari, Mansoor Shariff, Abdulkhaliq A. Alshadidi, Hassan A. Alhazmi, Asaad Khalid

**Affiliations:** 1Department of Prosthetic Dental Science, College of Dentistry, Jazan University, Jazan 45142, Saudi Arabia; 2Substance Abuse and Toxicology Research Center, Jazan 45142, Saudi Arabia; rshad100@yahoo.com (R.A.); hasalhazmi@gmail.com (H.A.A.); drasaad@gmail.com (A.K.); 3Prosthetic Department, College of Dentistry, King Khalid University, Abha 62529, Saudi Arabia; nmr.dnt@gmail.com (N.M.A.A.); mansoor_shariff@hotmail.com (M.S.); 4Applied Medical Sciences College, King Khalid University, Abha 61421, Saudi Arabia; aalshadidi@kku.edu.sa; 5Department of Pharmaceutical Chemistry, College of Pharmacy, Jazan University, Jazan 45142, Saudi Arabia; 6Medicinal and Aromatic Plants Research Institute, National Center for Research, Khartoum 11123, Sudan

**Keywords:** color measurements, khat, porcelain, surface roughness, natural teeth

## Abstract

*Background and Objectives*: Khat chewing is considered as a daily habit that is practiced by more than five million people globally. The effect of khat chewing on the surface roughness and the color stability of natural teeth and the material used in the fabrication of dental prosthesis remains unknown. This study was conducted to explore and compare the effect of khat homogenate (KH) on the surface roughness (Ra) and the average color changes (Δ*E**) amongst natural teeth and selected shades from different porcelain types, namely, feldspathic metal ceramic (MC) VM13, computer-aided design/computer assisted manufacture (CAD/CAM) feldspathic (Vitablocs Mark II), and multilayer zirconia (Ceramill Zolid PS) porcelains. *Materials and Methods*: Seventy samples were prepared from natural teeth, feldspathic MC, CAD/CAM Vitablocs Mark II, and zirconia porcelain. The Ra values were measured using a profilometer and expressed in micrometers, whereas the Δ*E** values were measured using VITA Easyshade^®^ V spectrophotometer for all samples before and after frequent immersion and thermocycling in KH for 30 days. The surface topography was used to assess the materials surfaces (glazed or polished) after KH immersion by using a white light interferometry machine. *Results*: Results revealed that the Ra and the Δ*E** values of the different types of tested porcelain were influenced by KH. The order of surface roughness values was glazed or polished MC > polished Zircon > polished Vitablocs Mark II > natural teeth. The lowest Δ*E** values were recorded for glazed Vitablocs Mark II and MC, and the values could be arranged as polished zircon > natural teeth > glazed zircon > polished MC > polished Vitablocs Mark II. P values were significantly varied (<0.001) among all the tested groups, except the zircon group (>0.05) for both Ra and Δ*E**. *Conclusions*: KH significantly affected both surface parameter and color of glazed or polished porcelain materials and natural teeth.

## 1. Introduction

Dental porcelain has become an extremely important material in prosthodontic dentistry because of its many advantages, such as biocompatibility, durability, long live survival, and excellent aesthetic capabilities with long-term follow up. This material shows translucency, color brightness, and intensity properties as natural non-stained teeth [1,2,3]. As a result of continuing improvements, several materials have been utilized to fabricate ceramic prosthesis, including feldspathic porcelain, glass-based ceramics, and zirconia-based ceramics [3,4]. In addition, for many years, MC feldspathic prostheses have been the most widely used restorative material in the aesthetic zone due to the clinical longevity and the accepted aesthetics of the restoration [5]. Nowadays, in modern dental practice, zircon-based ceramic is considered as one of the most popular materials to fabricate porcelain prostheses [6,7] due to its superior mechanical properties [7,8,9] and excellent biocompatibility compared with other dental ceramic [10,11,12]. 

Khat or qat plant (Catha edulis) is a natural central nervous system stimulant used by millions around the globe. It is cultivated in the south of Saudi Arabia [13]. Green khat leaves are chewed and kept in buccal vestibules as unilateral or bilateral in the form of bolus for at least 3–5 h [14,15]. It has an aromatic odor and an astringent and slightly sweet taste. Many different compounds are found in khat, including alkaloids, terpenoids, flavonoids, sterols, glycosides, tannins, amino acids, vitamins, and minerals. Phenylalkylamines and cathedulins are the major alkaloids and are structurally related to amphetamine [16]. Given the continuous contact of khat with oral tissues, it may affect hard and soft tissues, such as the salivary glands, leading to xerostomia and increasing caries rate [17]. Moreover, khat chewing leads to reduced salivary flow, increased viscosity, and lowered pH [18]. 

The stability of dental ceramics in the oral environment is directly related to high surface polish, subcritical crack propagation, and chemical inertness of material atoms, thereby enabling them to resist degradation in the oral environment [19]. Surface roughness deteriorates the biomechanical and the aesthetic value of dental restorations, increasing susceptibility to aging [20]. A rough-surfaced dental restoration will not only favor more plaque retention [21] and cause abrasive damage to opposing dentition [22], but it will also be weakened by the presence of surface flaws, which can cause material failure [7,23].

A spectrophotometer measures the reflectance or the transmittance of wavelengths from one object at a time without being affected by the subjective interferences of color [24,25], whereas a colorimeter measures light absorbance. The Commission Internationale de l’Eclairage L*a*b* system measures chromaticity and defines the color of an object in a uniform 3D space. The average color difference (ΔE*) is calculated for different materials and surface treatments using the following equation: ΔE* = ((L1* − L2*)^2^ + (a1* − a2*)^2^ + (b1* − b2*)^2^ “× ½ [26]. Motro et al. (2012) stated that if ΔE* = 0, the material tested is considered color stable, whereas values of ΔE* between zero and two suggest negligible color differences [27]. Alghazali et al. (2012) recommended that the color change values were compared with clinically acceptable and perceptible thresholds of 4.2 and 2.8, respectively [28]. The VITA Easyshade^®^ V spectrophotometer device is reported to be more accurate and precise than the other instruments for prosthetic material color measurements [29,30]. 

Resistance to staining of a prosthesis is an important clinical requirement for fracture resistance and other mechanical properties [31]. After a long period, inadequate color stability and staining may prompt a dentist to renew the restoration. The color stability of any prosthesis is affected by intrinsic factors caused by aging, extrinsic factors from surface staining of diet, plaque accumulation, surface staining absorption, and degradation agents [32]. 

A dental practitioner usually performs some intraoral occlusal adjustments to the porcelain covering materials. Such adjustments include removal of a part of the glazed layer from the ceramic surfaces, leaving some pores in the ceramic material, and creating a rough surface. These effects can cause the discoloration of the prostheses that requires either re-glazing or polishing to maintain luster and a stable color [27,33,34].

Khat chewing usually has a bitter taste and causes mouth dryness. To overcome these issues, large quantities of soft drinks or beverages and sugar tablets, which can lead to cervical discoloration in both enamel and dentine for a long period, are consumed and may lead to staining of teeth, attrition, and cervical caries at the chewing side [35,36]. This in vitro study aimed to derive and compare the effects of khat homogenate (KH) on the surface roughness and the color stability of the most commonly used brands of feldspathic metal ceramic (VM13), CAD/CAM feldspathic (Vitablocs Mark II) and CAD/CAM multilayer zirconia (Ceramill Zolid PS) porcelain materials. The null hypothesis of this study was that the surface roughness and the color stability of the tested dental porcelain (either glazed or polished) may not be affected by Khat chewing and thermocycling.

## 2. Materials and Methods

### 2.1. Study Design

In this in vitro study, 70 samples were included to assess and measure the effect of KH on the surface roughness, surface topography, and color changes of glazed or polished feldspathic metal ceramic, machinable feldspathic or multilayer CAD/CAM zircon porcelain, and natural teeth. Table 1, represented the materials and devices used in the study.

### 2.2. Sample Preparation and Fabrication

Seventy samples were prepared, fabricated, and divided into the four following groups.

There was the natural teeth group, which consisted of 10 extracted maxillary caries-free central incisors collected from surgical clinics. The teeth were polished with rubber cups with fluoride-free prophy paste and then stored in 10% aqueous formalin solution at room temperature Figure 1A. 

The group of the feldspathic metal ceramic VITA VM_(R)_13 consisted of 20 samples. These samples were constructed by a conventional method using 0.4 mm-thick green wax with dimensions of 10 mm × 10 mm. The MC samples were invested, burned out, and casted with nickel-chromium dental casting alloy (Wiron^(R)^ 99, Bego, Germany) following the manufacturer’s instructions. The metal samples were cut from their sprues, leaving 2 mm to facilitate handling of the samples during packing of different porcelain layers. The metal samples were sandblasted with aluminum oxide particle to remove the excess oxide layer. A feldspathic ceramic VITA VM_(R)_13 (VITA, Zahnfabrik, Germany) was used for porcelain build-up. The opaque layer on the metal substrate had a thickness of 0.3 ± 0.1 mm, while the body porcelain had a thickness of 2.0 ± 0.3 mm [37]. The feldspathic MC samples were packed and glazed following the manufacturer’s recommendations Figure 1B. 

In the group of machinable feldspathic porcelain Vitablocks Mark II (Vita Zahnfabrik, Germany), the samples were prepared by installing the blocks in a milling CAD/CAM machine (Amann Girrbach, Germany) to produce 20 discs with dimensions of 10 mm × 10 mm and thickness of 2.0 ± 0.3 mm Figure 2A. All samples were glazed in accordance with the manufacturer’s recommendations. 

For the group of pre-sintered zirconia, Ceramill Zolid multilayer PS (preshaded) blocks were used to fabricate the samples by installing them in the milling CAD/CAM machine (Amann Girrbach, Germany) to slice 20 disc-shaped samples with dimensions of 10 mm × 10 mm and thickness of 2.0 ± 0.3 mm. All samples were crystallized and sintered following the manufacturer’s recommendations Figure 2B. The shade B1 was used for all the porcelain veneering materials, and it is the most selected shade in daily practice for prosthesis fabrication and construction. 

### 2.3. Surface Treatments of Samples

The 70 prepared samples of the four groups were divided into two equal subgroups with 35 samples each. The natural teeth samples comprised the control group with five samples for each subgroup, and measurements were carried out without any surface treatments. The first subgroup samples (10 samples from each group) were obtained from the laboratory with glazed surfaces without further treatment, whereas the samples in the second subgroup (10 samples from each group) were polished with a porcelain polishing kit (Figure 3) to represent the clinical situation for the prostheses by using the protocol recommended by the company. The sequential polishing of every 10 samples from each ceramic material was done with polishing burs at equal number of grinding at one direction at constant speed and under control pressure at 5 ± 0.25 kg [37].

### 2.4. Surface Roughness Measurements

All samples were tested using a profilometer device (Surface Roughness Tester, Perthometer M2, Mahr GmbH, Germany). The surface roughness was expressed in terms of roughness average (Ra), which is usually measured in micrometer (μm). For each sample, two readings were registered, and the mean values were calculated before immersion in KH and thermocycling. To measure the surface roughness of each sample, the stylus was moved above the surface of the samples twice in three different directions around the center of the sample for a distance of 4 mm according to the profilometer’s instructions to record all peaks and recesses that characterized the surface. This was considered as Ra value before treatment. Ra measurements followed the ISO 11562 recommendations to standardize gaining of the results [38].

### 2.5. Color Measurements 

Color was measured for all assigned samples by the same operator under the same settings and gray background. Color measurements were made using Vita Easy shade probe spectrophotometer (VITA Easyshade^®^ V, VITA, Germany). The spectrophotometer was used to measure the Commission International D’ Eclairage CIE-Lab values of the samples to provide a numerical representation of 3D color measurements. These measurements were previously used in studies assessing the shades of both porcelain and teeth. *L**, *a**, and *b** were measured twice, and the mean value was obtained. The means of color data with the standard deviations of tooth surfaces were calculated as previously described [28,33]. This value was considered the average color Δ*E** before KH immersion using the equation mentioned earlier. 

### 2.6. Sample Immersion and Thermocycling

KH was prepared and offered by the Substance Abuse and Toxicology Research Center, Jazan University (Figure 4). The homogenate of khat leaves was prepared from fresh mincing fresh khat leaves in 100% of distilled water (V/W) and finely minced. The KH was then kept in a −80 °C ultra-low temperature freezer until use. KH was then mixed with NaOH until its pH was similar to the pH of saliva and the oral cavity. All samples were immersed in KH for 30 days as mentioned in previous in vitro studies [39,40,41]. The same procedures were executed daily to obtain fresh solutions. During the immersion time, an aging process was conducted using a thermocycling machine, where 100 cycles were accomplished every day in 5 °C cold water and then in 55 °C hot water daily (3000 cycles). All samples were dipped for 10 times in distilled water following removal from immersion media. Samples were wiped dry with tissue paper and left in place for complete dryness. 

After 30 days, the surface roughness and the colors of the samples were measured after KH immersion and thermocycling with the same profilometer and VITA Easyshade^®^ V spectrophotometer devices, and the readings were registered. These readings were recorded as Ra and the average color changes (Δ*E**) values after immersion and thermocycling. All procedures for sample preparation, fabrication, finishing and polishing, surface and color measurements, sample immersion, and thermocycling were carried out by the same operator. The Ra values indicate the average of surface roughness before and after immersion, while Δ*E** values refer to the average color changes, which were calculated by the aforementioned equation.

### 2.7. Surface Evaluation and Qualitative Analysis 

A sample from each type of porcelain in glazed or polished surface was scanned after KH immersion and thermocycling. The surface topography of six samples was represented graphically via white light interferometry (Contour GT-K1, Bruker Nano GmbH, Berlin, Germany) under 50× magnification with back scan and length parameters of 20 µm in VSI/VXI mode to obtain a 3D rendering of the sample surfaces. Vision 64 software (Bruker Nano GmbH, Berlin, Germany), which is part of the GT-K1 system, was used to copy the surface topography parameters.

### 2.8. Statistical Analysis

The average Ra values and the mean color changes Δ*E** differences of natural teeth and porcelain sample restorations (VM_(R)_13, Vitablocs Mark II, Ceramill Zolid) in terms of glazing and polishing were recorded and then compared before and after KH immersion. Microsoft Excel 13 software was used to input the data, which were analyzed using Statistical Package for Social Science (SPSS) version 22.0 (SPSS Inc., Chicago IL, USA). ANOVA test followed by Bonferroni test were conducted to detect any significant difference between and within the groups at *p* > 0.05.

During statistical analysis, the 70 samples were represented by letters from a to g, in which (a) represented natural teeth before and after KH immersion and thermocycling; (b–c, d–e, and f–g) were used for glazed and polished MC, Vitablocs Mark II, and zircon, respectively. 

## 3. Results

In this in vitro study, 70 samples were included. Ra is the average of surface roughness before and after immersion in KH and thermocycling. In general, the Ra values for the tested materials were as follows: natural teeth > polished zircon > polished Vitablocs Mark II > glazed zircon > glazed Vitablocs Mark II > glazed and polished MC. Comparing the recorded Ra values of glazed or polished samples before immersion, the effect of khat was obvious, particularly on the polished samples with significant differences between the group of all ceramic with (*p* < 0.05, Figure 5). The Ra values significantly varied between natural teeth and all tested glazed or polished porcelain, except between natural teeth and polished zircon (*p* < 0.05, Table 2). The mean values of Δ*E** were calculated by the aforementioned equation. The Δ*E** values could be arranged as follows: natural teeth > zircon polished or glazed > polished MC > polished Vitablocs Mark II > glazed Vitablocs Mark II and MC. The Δ*E** values significantly differed between natural teeth and all tested glazed or polished porcelain (*p* > 0.05, Table 3 and Table 4). The surface of natural teeth was polished before immersion for staining removal only; therefore, Ra and natural teeth (in Table 2 and Table 3) refer to values before and after immersion. 

The representative white light interferometer microscope images of the three tested materials after KH immersion and thermocycling of the selected shade are shown in Figure 6 A–F. The red areas represent the part of the surface with the highest height (the peaks), while the blue areas represent the part of the surface with the lowest height (the valleys). All the six tested samples showed a high variable pattern of peaks and valleys across the examined surfaces. These patterns were not identical across each surface in either glazed or polished surfaces. Moreover, microscopic images for MC samples shown in Figure 6A,B present a non-uniform surface with distinct sharp projections dotted with pore. However, Mark II showed a slightly smooth surface with randomly located pores. A low profile, which is interrupted by randomly located rounded-off projections, characterized the material surface Figure 6C,D. Zirconia demonstrated a highly irregular surface pattern with heights and valleys—wide, irregular, and deep scratch areas were shown across the surfaces (Figure 6E,F). 

## 4. Discussion

Based on the available information about the khat association with many reported health, social, and economic problems affecting the consumers and their families in a number of African, Asian, European, and North American countries, the WHO Expert Committee on Specifications for Pharmaceutical Preparations (Thirty-third report, 1993) has recommended that khat should be considered as a controlled drug. 

This in-vitro study was designed to assess the effect of KH immersion with thermocycling on the surface roughness and the color stability of the most used types of porcelain materials in the fabrication of the dental prosthesis. The surface roughness of glazed surfaces was compared with that of polished porcelain surfaces using a manual polishing kit (Figure 3). The present study is the first to study the effect of KH on glazed and polished porcelain and natural teeth. The overall results of the current study complement a wide range of adverse effects of KH on porcelain samples in both forms and natural teeth. In addition, the null hypothesis of this study was rejected, because the surface roughness and the color stability of tested natural teeth and glazed or polished porcelain materials were affected by KH immersion and thermocycling. 

Few studies in relation to KH and dental materials have been conducted. An in-vitro study investigated the effect of khat extract on the color of composite materials [39]. They concluded that khat extract alone or in combination with beverage drinking shows the highest effects on the color stability of different types of recent composite resins, and the color changes are clinically perceptible. Our previous in vivo study showed that khat chewing significantly affects oral biofilm formation in the presence of different filling and porcelain materials, and it may lead to changes in dental color restorations and prostheses [42]. Khat chewing was reported to be associated with demineralization of composite restorative materials at the composite tooth interface, thereby resulting in altered color of the composite and the tooth structure [18].

The smoothness, and the surface quality of the prosthetic material are important to achieve the desired aesthetic appearance and long-term clinical success [43], because rough surfaces affect discoloration and shade matching. Alp and Subaşı (2019) and Al-Hebshi et al. (2005) assessed the effect of khat on salivary parameters and found that it caused inadequate and decreased salivary flow rate with high viscosity and lowered the pH of saliva. These factors were associated with increased risk of teeth discoloration [44,45]. 

The Ra value was used as a roughness parameter in the current study because it is the most commonly used parameter for surface assessment, thereby facilitating a convenient comparison with other studies [38]. The effect of KH was obvious in all samples to different degrees (Figure 6). Moreover, Table 2 shows that the value was the highest on natural teeth (4.91 μm). This result was in agreement with a previous study, which reported that natural teeth become more discolored due to mechanical and acidic effects of KH [46]. Khat was reported to alter the enamel surface of teeth, which may collect more beverages on the surface, resulting in the discoloration of the tooth after a prolonged period [14]. However, Lawson et al. recorded an Ra value of 2.63 ± 1.14 μm for enamel; this value was lower than the obtained values in the present investigation [47]. In this study, the lowest Ra value was obtained from glazed or polished MC samples (0.83 and 1.18 μm), followed by samples of glazed Vitablocs Mark II (1.26 μm). This finding was in agreement with another study, which concluded that feldspathic and low fusing porcelain have stable surfaces [48,49,50]. The highest mean Ra was gained from zircon polished samples (2.23 μm), which was consistent with that reported for polished porcelain samples [51,52]. Highly polished zirconia is more desirable than other porcelain materials because of its clinical behavior, chemical composition, and properties [51,53]. A previous study reported an Ra value of (2.73 ± 1.49 μm) for their tested porcelain samples [47]. In addition, the surfaces of zirconia samples may undergo some changes because of aging and short-term storage, which was obvious in the zirconia samples in the present study [40,54].

The CIE Lab system was chosen in the current study for assessment of chromaticity and to evaluate the average color differences, because it is well suited for the determination of small color differences [28,50]. The average color differences were assessed in terms of perceptibility and acceptability for small color differences, because of their role as a guide control during the selection of porcelain materials for khat chewers. The Δ*E** measurements of VITA Easyshade^®^ V spectrophotometer devices are more precise and accurate than those of other digital instruments [29,30]. 

The effect of different beverages on the ΔE* of porcelain materials was described in the literature. Acar et al. (2016), Saba et al. (2017), Colombo et al. (2017), Alp et al. (2018), and Sarikaya and Guler (2011) evaluated the effect of coffee [31,40,45,48,50], while Sarıkaya et al. (2018) and Colombo et al. (2017) assessed the effect of Coca-Cola [40,55]. On the other hand, a single study carried out by Alghazali et al. (2019) [33] investigated the effect of Arabic coffee, and Yilmaz et al. (2006) inspected the effect of methylene blue [56]. These studies examined the effect of beverages on polished or glazed porcelain materials. All the used immersion beverages with thermocycling had an effect on different types of porcelain samples, and mean color changes Δ*E** were higher in polished samples than in glazed samples. This result supported our findings, even though we used KH immersion, because khat could have an acidic effect on the teeth, resulting in a discoloration effect of KH on teeth surfaces after a certain period of chewing. KH has the fibrous nature of khat leaves; thus, it may cause some mechanical alterations in porcelain surfaces due to the friction mechanism and the hardness of khat [57,58]. However, the effect of khat on surface roughness, color of natural teeth, and various porcelain materials needs to be further explored. In this study, the value of the Δ*E** (2.12) was slightly high (Table 3), and this can be explained by the high degree of fluorosis among the population in Jazan city. Alabdulaaly et al. concluded that the fluoride levels in Jazan exceeded the maximum allowed limits [59] 

Algazali et al. performed a clinical in vivo study and found that Δ*E** of 2.8 is clinically acceptable [28]. Al Ghazali et al. (2019), Maciel et al. (2019), and Sarikaya and Gular (2011) concluded that Δ*E** values of ≥1 to ≤3.3 can be detected by the eyes and are clinically acceptable [33,50,60]. These values coincide with the values obtained in this study, in which Δ*E** ranged from 0.28 for glazed samples of Vitablocs Mark II to 2.25 for polished samples of zirconia porcelain (Table 3). Similarly, Palla et al. (2018) and Maciel et al. (2019) reported Δ*E** of 3.7 for all Ivoclar-Vivadent Pressable Ceramic (IPM) e.max-tested samples immersed in different beverages with thermocycling [60,61]. This result was close to the value obtained in this study for natural teeth (3.45) (Table 3). Overall, glazed and polished porcelain materials showed Δ*E** values of 0.23–2.52 units, except for samples of natural teeth after (3.45) KH immersion and thermocycling. Therefore, the behavior of all tested samples could be considered acceptable. 

A group of in vitro studies was carried out by Lee et al. to assess the effect of different types of mouth wash in relation to different type of feldspathic and zirconia CAD/CAM ceramic materials [61,62]. They concluded that significant differences were observed in the average color change Δ*E** values of monolithic zirconia as well as the feldspathic ceramic materials group; the ceramic became brighter, opaquer, less glossy, and rougher after rinsing with the whitening mouthwash. Additionally, long-term use of specific mouth wash may cause deterioration of optical and surface properties of high-translucency CAD/CAM prosthetic materials. Lee et al. concluded that ultrasonic scaling resulted in significant changes in the surface roughness and showed scrapes as well as surface deterioration of the Vitablocs Mark II ceramic materials after scaling [63]. The results of the current study demonstrated a prominent effect of KH on the Ra as well as the Δ*E** values on the same materials.

Intraoral restoration surfaces can be prepared using various glazing and mechanical polishing techniques. The techniques recommended by manufacturers involve the use of different glazing procedures combined with crystallization or after crystallization and polishing for all porcelain to maintain and obtain color stability, translucency, and surface roughness [55,56,60]. A strong and direct correlation was observed between Ra and ΔE* of all tested ceramics. This correlation was obvious in our samples after crystallization and polishing, which resulted in a clinically acceptable color change difference after KH immersion and thermocycling. The only exception was observed in natural teeth samples, and it could be associated with the mechanical effect of KH on natural teeth. 

Limited studies have used white light interferometer microscope images to examine the surface topography of both traditional and recent ceramic materials after application or immersion of any beverage or stained materials. In this study, the polished samples of zirconia (Figure 6F) showed high surface topography alteration compared with glazed samples (Figure 6E), indicating that the mechanical and the acidic effects of KH were obvious. KH is fibrous in nature, and this property may cause a mechanical effect on the surfaces of porcelain, especially those of polished samples [58]. The glazed surfaces of feldspathic MC and CAD/CAM Vitablocs Marke II porcelain samples reflected the lowest effect of KH on their surface topography (Figure 6A,C). All the changes on the surface topography were reflected in the obtained Ra values, resembling equal effects on the degree of Ra shown in Table 2. The images obtained in this study were unlike the images documented in a previous report [64]. The differences could be due to the varying components and the surface characteristics of the examined samples used and the laboratory techniques adopted for the fabrication of the tested samples in both studies. 

Considering that this study was the first to analyze the effect of KH on different types of glazed or polished porcelain and natural teeth, it involved some limitations. First, the current study was an in vitro study and allowed staining on both sides of the material. In clinical situations, the material is cemented to a tooth structure and is exposed to KH and light on the outer surface only. Moreover, the color of the samples is a combination of gray background and the sample’s color, and the average color values may change in oral cavity where different backgrounds exist. The results of this study should be simulated in further clinical studies to mimic the effect of actual khat chewing, which is usually associated with soft drinks and smoking on Ra and Δ*E** of other brands of dental porcelain.

## 5. Conclusions

This in vitro study shows that KH exerts significant effects on the overall surface roughness of the examined sample surfaces. The effects were higher on natural teeth and zirconia than on feldspathic porcelain, whether manually packed or machinable by CAD/CAM. KH showed acceptable changes in Δ*E** for both glazed or polished porcelain materials and natural teeth with significant differences among all groups and subgroups, except between natural teeth and zircon in terms of Ra measurements. The limitations of this study could be addressed by further clinical studies to mimic the effect of khat chewing habit on Ra and Δ*E** of different brands and recent types of dental porcelain materials.

## Figures and Tables

**Figure 1 medicina-56-00234-f001:**
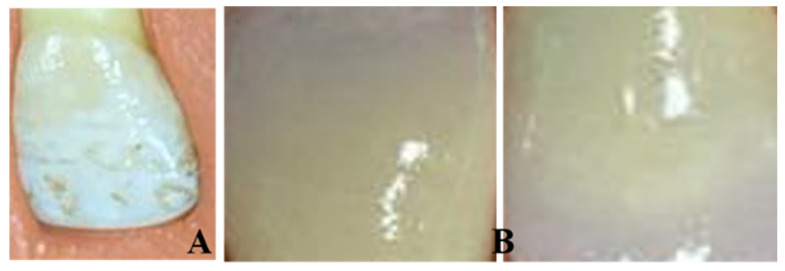
Natural teeth (**A**) and feldspathic metal ceramic (MC) (**B**) samples.

**Figure 2 medicina-56-00234-f002:**
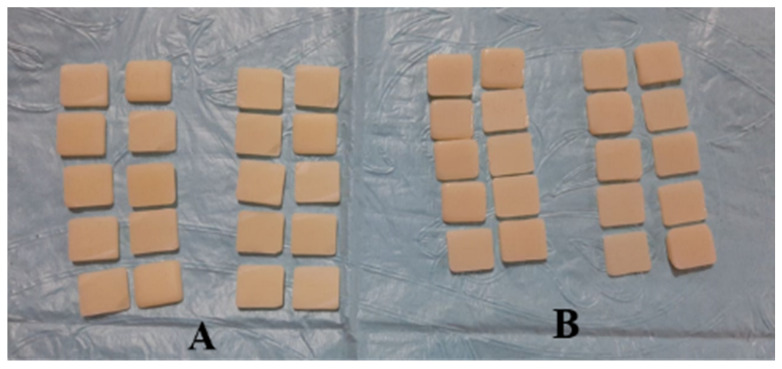
CAD/CAM Vitablocks Mark II (**A**) and Zircon (**B**) samples.

**Figure 3 medicina-56-00234-f003:**
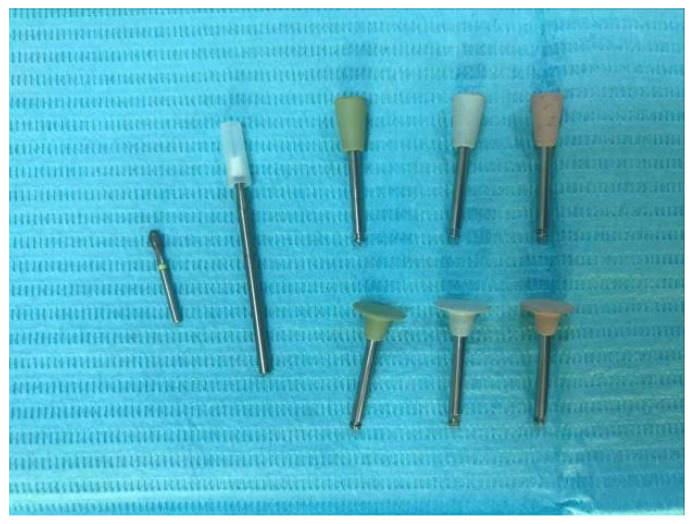
Polishing kit used in the study.

**Figure 4 medicina-56-00234-f004:**
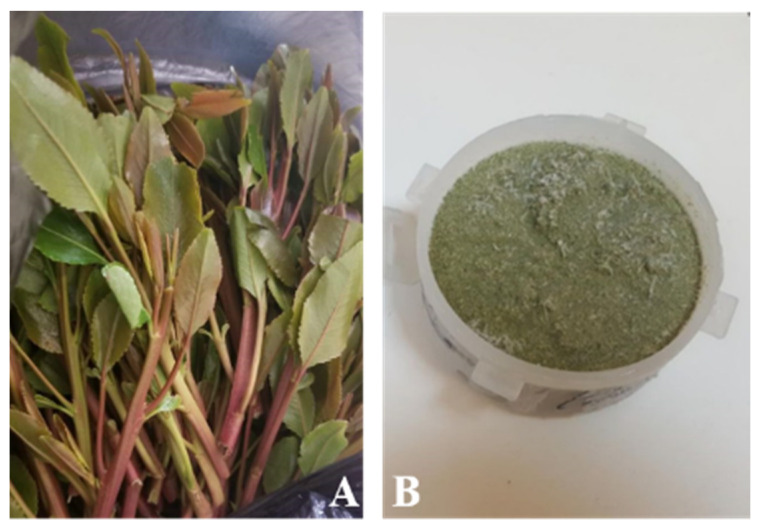
Khat extract used in the study.

**Figure 5 medicina-56-00234-f005:**
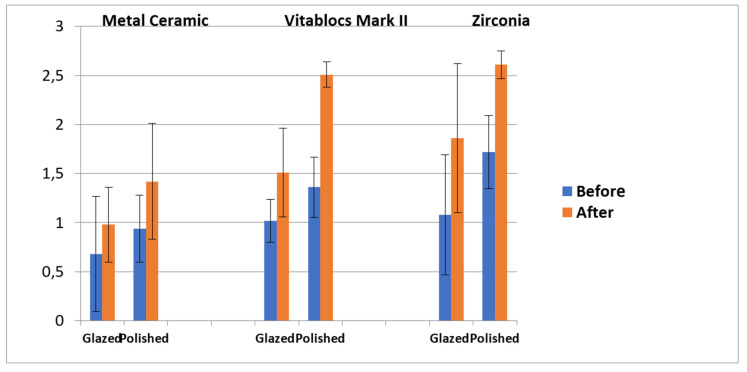
Mean and standard deviation of surface roughness (Ra) before and after KH immersion.

**Figure 6 medicina-56-00234-f006:**
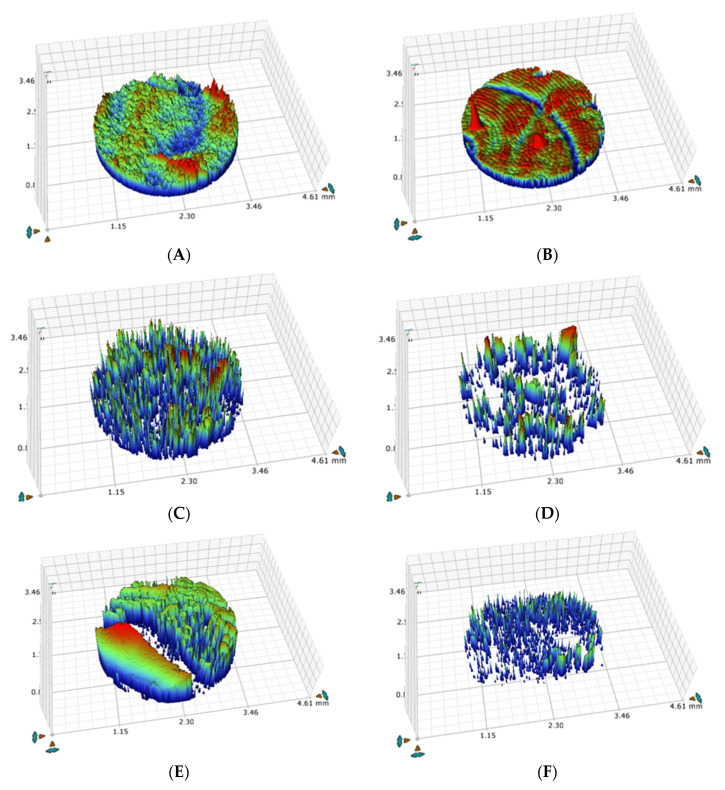
(**A**–**F**). Representative images of white light interferometer microscope of tested porcelain samples obtained at 50× magnification. MC, (**A**,**B**); Mark II, (**C**,**D**); and Zirconia, (**E**,**F**).

**Table 1 medicina-56-00234-t001:** Materials and devices used in the study.

Material/Device Type	Type (Brand Name)	Composition	Manufacturers	Lot #	Shades
Nickel chromium casting alloy	Wiron^(R)^ 99	Ni 65%, Cr 22.5%, Mo 9.5%, Nb 1% Si 1%, Fe 0.5%, Ce 0.5%, C_max_ 0.021	BEGO, Germany	9868	Silver
Feldspathic CAD/CAM porcelain blocks,	Vitablocs Mark II	Fine-particle feldspar glass ceramic, low-to-moderate <50% leucite-containing.	VITA Zahnfabrik, Bad Säckingen, Germany	42650	B1
Feldspathic P/L porcelain body	VITA VM_(R)_13	Silicon dioxide and possess a glassy matrix, assorted quantities of potassium, sodium, barium, or calcium	VITA Zahnfabrik, Bad Säckingen, Germany	49510	B1
Zircon CAD/CAM	Ceramill Zolid multilayer PS	ZrO2 + HfO2 + Y2O3: ≥99.0, Y2O3: 8.5–9.5, HfO2: ≤5, Al2O3: ≤0.5, Other oxides: ≤1	Amann Girrbach, Germany	32976-FB	B1 Light`
Khat or Qat	*Catha edulis* plant	Alkaloids, terpenoids, flavonoids, sterols, glycosides, tannins, amino acids, vitamins, minerals.			
Surface Roughness Tester	Profilometer	Device recorded graphically the average height of profile above and below a center line along the given length of a sample	Perthometer M2, Mahr GmbH, Germany	NHT-6	
Spectrophotometer	VITA Easyshade^®^ V	Device used to measure wavelength transmitted from one object at a time, without being affected by subjective interferences of color	VITA Zahnfabrik H. Rauter GmbH & Co. KG, Bad Sackingen, Germany	10180	
Surface roughness and topography tester	White Light Interferometry Microscope	3D printer of surface characteristics	Contour GT-K1, Bruker Nano GmbH, Berlin, Germany		

CAD/CAM: computer-aided design/computer assisted manufacture.

**Table 2 medicina-56-00234-t002:** Mean and standard deviation of surface roughness (Ra) in each tested group.

Tested Material	Mean Ra ± SD – Glazed	Mean Ra ± SD – Polished	*p* Values
MC	0.83 (0.266) ^a,c,d,e,f,g^	1.18 (0.127) ^a,b,d,e,f,g^	<0.001
Vitablocs Mark II	1.26 (0.274) ^a,b,c,e,f,g^	1.93 (0.306) ^a,b,c,d,f,g^	<0.001
Zircon	1.32 (0.116) ^b,c,d,e,g^	2.23 (0.114) ^b,c,d,e,f^	<0.001

Different superscript letters indicate statistically significant difference inside the respective subgroup (*p* < 0.05) based on ANOVA followed by Bonferroni tests.

**Table 3 medicina-56-00234-t003:** Mean and standard deviation of the average color (Δ*E**) parameter in each tested group.

Material Type	Mean ± (SD) – Glazed	Mean ± SD – Polished	*p* Values
MC	0.56 (0.080) _a,c,d,e,f,g_	1.34 (0.126) _a,b,d,e,f,g_	<0.001
Vitablocs Mark II	0.28 (0.079) _a,b,c,e,f,g_	0.73 (0.134) _a,b,c,d,f,g_	<0.001
Zircon	1.58 (0.141) _a,b,c,d,e,g_	2.52 (0.188) _a,b,c,d,e,f_	<0.001

Different subscript letters indicate statistically significant difference inside the respective subgroup (*p* < 0.05) based on ANOVA followed by Bonferroni tests.

**Table 4 medicina-56-00234-t004:** Mean and standard deviation of surface roughness (Ra) and average color change (Δ*E**) for Natural teeth group.

Parameter	Before Immersion *	After Immersion *	*p* Values
Surface roughness (Ra)	2.75 (0.213)	4.91 (0.338)	<0.001
Average color (Δ*E**)	2.12 (0.201)	3.45 (.411)	<0.001

* Mean ± SD before and after immersion in KH. Statistically significant difference inside the respective subgroup (*p* < 0.05) based on ANOVA followed by Bonferroni tests.

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
