# Peer review of "Effects of Khat on Surface Roughness and Color of Feldspathic and Zirconia Porcelain Materials under Simulated Oral Cavity Conditions"

_medicina, 2020, doi:10.3390/medicina56050234_

Round 1

Reviewer 1 Report

Impact: Moderate
Methods: Appropriate

The aim of this in vitro study is to evaluate and compare the effects of KH (Catha edulis) on the surface roughness and color stability of the most commonly used brands of feldspathic metal-ceramic (VM13), CAD/CAM feldspathic (Mark II), and CAD/CAM multilayer zirconia (Ceramill Zolid PS) porcelain materials. The quality of the research is adequate, the research is well conducted, and the manuscript is well written. However, the topic is limited for some Arabic countries because the KH is considered a drug by OMS. This observation must write and comment in the discussion.

line 48: The authors could enter a more recent reference as:
"Arena A, Prete F, Rambaldi E, Bignozzi MC, Monaco C, Di Fiore A, Chevalier J. Nanostructured Zirconia-Based Ceramics and Composites in Dentistry: A State-of-the-Art Review. Nanomaterials (Basel). 2019 Sep 29;9(10)."

line 346-256: The authors must specify the section "Authors Contributions"

Author Response

Dear Sir,

We would like to thank you very much for your very valuable comments. Please find below point to point answers for all comments:

  1. However, the topic is limited for some Arabic countries because the KH is considered a drug by OMS. This observation must write and comment in the discussion.

Answer: The suggested information is now added to discussion (Line 240-243 )

  1. line 48: The authors could enter a more recent reference as: 
    "Arena A, Prete F, Rambaldi E, Bignozzi MC, Monaco C, Di Fiore A, Chevalier J. Nanostructured Zirconia-Based Ceramics and Composites in Dentistry: A State-of-the-Art Review. Nanomaterials (Basel). 2019 Sep 29;9(10)."

Answer: Three recent references are now added to the manuscript Ref 8-9 and 10

  1. line 346-256: The authors must specify the section "Authors Contributions"

Answer: Done, and it is now mentioned at the end of the manuscript.  

Reviewer 2 Report

First of all, congratulate you on your choice. This work makes it possible to draw good advice for the daily work of dentists.
Thus, and since I have some experience in the area, I would like to make a constructive criticism of your work.

Regarding the methodology, where you write: "whereas the samples in the second subgroup (10 samples from each group) were polished with a porcelain polishing kit. Every 10 samples from each ceramic material were polished using a polishing kit with equal number of grinding in one direction under a constant load. ":

1 - How did you manage to standardize the polishing throughout the sample? As described, it is completely impossible to replicate the polishing in all samples and thus have reproducibility.

2 - How did you achieve constant force during the sample polishing process? It is not indicated how you did it.

3 - I suggest you polish with the sequential Grinding papers with different grit at a constant speed.

Regarding the statistical analysis: "ANOVA and paired Student’s t-test, followed by Bonferroni test, were conducted to detect any significant difference between and within the groups at P> 0.05.":

1 - The statistical tests used are not correct. It is not correct to use two parametric tests in this situation.

 2 - I suggest carrying out a 2-way repeated measures ANOVA test, which seems more appropriate.

Results:
The following sentence is not a result but a part of the methodology: "In this in-vitro study, 70 samples were included (represented by letters from a to g), in which a represent natural teeth before and after KH immersion and thermocycling; b –C, d – e, and f – g were used for glazed and polished MC, Mark II, and zircon, respectively. "

Author Contributions are not properly formatted as provided in the guidelines

Author Response

Dear Sir,

We would like to thank you very much for your very valuable comments. Please find below point to point answers for all comments:

.

  1. Regarding the methodology, where you write: "whereas the samples in the second subgroup (10 samples from each group) were polished with a porcelain polishing kit. Every 10 samples from each ceramic material were polished using a polishing kit with equal number of grinding in one direction under a constant load.

 - How did you manage to standardize the polishing throughout the sample? As described, it is completely impossible to replicate the polishing in all samples and thus have reproducibility.

 - How did you achieve constant force during the sample polishing process? It is not indicated how you did it.

Answer: The sample were placed in a laboratory balance and the manual pressure applied during polishing was fixed at 5 ± 0.25 kg

- I suggest you polish with the sequential Grinding papers with different grit at a constant speed.

Answer: The sequential polishing steps were done with polishing burs at equal number of grinding at one direction at constant speed and under control pressure at 5 ± 0.25 kg. This is now stated with reference in the manuscript (Line 144-146)

  1. Regarding the statistical analysis: "ANOVA and paired Student’s t-test, followed by Bonferroni test, were conducted to detect any significant difference between and within the groups at P> 0.05.":

 - The statistical tests used are not correct. It is not correct to use two parametric tests in this situation.

  - I suggest carrying out a 2-way repeated measures ANOVA test, which seems more appropriate.

Answer: As suggested by the reviewer, paired Student’s t-test is now removed from the manuscript keeping ANOVA test.

  1. The following sentence is not a result but a part of the methodology: "In this in-vitro study, 70 samples were included (represented by letters from a to g), in which a represent natural teeth before and after KH immersion and thermocycling; b –C, d – e, and f – g were used for glazed and polished MC, Mark II, and zircon, respectively. "

Answer: As suggested the sentence is now moved to the methodology section (statistical analysis).

  1. Author Contributions are not properly formatted as provided in the guidelines.

Answer: Author Contributions is now properly formatted as provided in the guidelines.

Reviewer 3 Report

  1. According to the abstract, only 5 million people chew knat worldwide. Most readers of international journals will not be interested in it. It would be more appropriate to submit this manuscript to local journals than to international journals.

  1. Do people chew knat with anterior teeth? If people chew knat with posterior teeth and put it in the posterior buccal vestibules, it will mainly affect the posterior teeth. However, it is very rare that the posterior teeth is restored with the weak feldspathic porcelain material, and the color of the posterior teeth is less important. So I don't quite understand why the authors evaluated the color changes of the feldspathic porcelain material by knat.

  1. The Vita Easyshade device has various versions. The authors should describe the version of the device used in the study (Table 1).

  1. (Table 1) Vitblocs Mark II is the correct name, not Vitablock Mark II.

  1. Most studies published today use ΔE00 fomula when evaluating color changes or differences. Is there a reason to use ΔE instead of ΔE00?

  1. I do not understand the contents of Table 3. How can the color changes ΔE be divided between ‘before’ and ‘after’? I think the changes between 'before' and 'after' should be the ΔE value. What is 'Glazed' in the 'Before' column and 'Polished' in the 'After' column? Confusing.

Author Response

Dear Sir,

We would like to thank you very much for your very valuable comments. Please find below point to point answers for all comments:

  1. According to the abstract, only 5 million people chew khat worldwide. Most readers of international journals will not be interested in it. It would be more appropriate to submit this manuscript to local journals than to international journals.

Answer: It is true that khat chewing is a profound social habit in many Arabian and African countries, However khat is also a universally growing problem as khat chewing is spreading in many African, Asian, European and North American countries.

  1. Do people chew knat with anterior teeth? If people chew knat with posterior teeth and put it in the posterior buccal vestibules, it will mainly affect the posterior teeth. However, it is very rare that the posterior teeth is restored with the weak feldspathic porcelain material, and the color of the posterior teeth is less important. So I don't quite understand why the authors evaluated the color changes of the feldspathic porcelain material by knat.

Answer: During chewing, khat is usually crushed properly in between the anterior teeth, then it is placed in the buccal vestibule which may extend to the anterior teeth of the chewing side.  Feldspathic porcelain can be used with metal of all-ceramic crowns or bridge as veneering material. There is growing interest in the aesthetic restoration.

 The Vita Easyshade device has various versions. The authors should describe the version of the device used in the study (Table 1).

Answer: The version of the device used in the study is now mentioned as requested in Table 1.

 (Table 1) Vitblocs Mark II is the correct name, not Vitablock Mark II.

Answer: The name is now corrected as requested 

 Most studies published today use ΔE00 formula when evaluating color changes or differences. Is there a reason to use ΔE instead of ΔE00?

Answer: We have no reason not to use ΔE00, However, ΔE was used following a recent article published in 2019 your esteemed journal (Reference # 9) 

 I do not understand the contents of Table 3. How can the color changes ΔE be divided between ‘before’ and ‘after’? I think the changes between 'before' and 'after' should be the ΔE value. What is 'Glazed' in the 'Before' column and 'Polished' in the 'After' column? Confusing.

Answer: The confusion descriptions are now clarified in Table 2 and Table 3

Round 2

Reviewer 2 Report

Nothing to report

Author Response

We would like to express our gratitude for your highly appreciated comments

Reviewer 3 Report

Thank you for revising your manuscript. However it still has serious flaws.

  1. Tables 2 and 3 still have serious flaws.

1) Authors did not glazed natural teeth, but Tables 2 and 3 appear to authors glazed natural teeth. It is not enough to prevent misunderstandings by simply denoting asterixes.

2) Line 186 and Table 2)

ΔE means a change or difference in color, but Ra does not mean a change or difference, but only the state.

If the values of MC, Mark II, and Zircon in Table 2 are the mean values of the Ra before and after immersion, it cannot show the effect of khat treatment. As you know, mean value indicate neither a change nor difference.  

How can readers know the effects of khat on the surface roughness in Table 2 and 1st paragraph of the Results section?

You only showed that the surface roughness values of the restorative materials are different from that of the natural teeth, there is no mention of the effect of khat.

3) In Table 3, ‘Natural teeth group’ has two ΔE results of 2.12 (0.201) and 3.45 (.411).

The ΔE results of 2.12 (0.201) is for control group? If it is for control group, ‘before immersion (Line 220)’ is not correct expression, because the control group will not be immersed.

In addition, why is the ΔE value of the control group of natural teeth so high as 2.12?

4) The p value 0.000 should be expressed as <0.001.

5)  ‘P within group’ is a strange expression. What group do you mean? You have many kinds of groups including subgroups. Readers will be confused.

  1. Line 126) All samples were glazed? Including polishing group? Glazed and then polished?

Line 133 and 140) Did the polishing group of zirconia also get glazing and then polished?

  1. You should revise not only Table 1 but also your entire manuscript regarding the Vitablocs Mark II and Vita Easyshade V.

  1. Line 96 and 247) I think the null hypothesis should be reversed and rejected based on the p-values you expressed in footnotes of Tables 2 and 3.

  1. Line 214) ‘Significantly differed’ and ‘P>0.05’ ?

Foot notes of Table 2 and 3: Different superscript letters indicate statistically significant difference inside the respective subgroup (P < 0.05)

  1. Line 131) CAC/CAM --> CAD/CAM

  1. Finally I suggest to refer and insert the following recent papers regarding the evaluation of color and roughness changes of restorative materials:

1) Optical and Surface Properties of Monolithic Zirconia after Simulated Toothbrushing

https://doi.org/10.3390/ma12071158

2) Effects of ultrasonic scaling on the optical properties and surface characteristics of highly translucent CAD/CAM ceramic restorative materials

https://doi.org/10.1016/j.ceramint.2019.04.177

3) Colour stability and surface properties of high‐translucency restorative materials for digital dentistry after simulated oral rinsing

https://doi.org/10.1111/eos.12676

Author Response

Answers to Reviewer 3 (Round 2)

Comment:

Tables 2 and 3 still have serious flaws.

 Authors did not glazed natural teeth, but Tables 2 and 3 appear to authors glazed natural teeth. It is not enough to prevent misunderstandings by simply denoting asterixes.

Answer:

Results of natural teeth is now separated in new table (Table-4). It has also been clarified in an additional statement in the results section (Line: 218-222)

Comment:

 Line 186 and Table 2)

ΔE means a change or difference in color, but Ra does not mean a change or difference, but only the state.

If the values of MC, Mark II, and Zircon in Table 2 are the mean values of the Ra before and after immersion, it cannot show the effect of khat treatment. As you know, mean value indicate neither a change nor difference.  

How can readers know the effects of khat on the surface roughness in Table 2 and 1st paragraph of the Results section?

You only showed that the surface roughness values of the restorative materials are different from that of the natural teeth, there is no mention of the effect of khat.

Answer:

As discussed in the manuscript, the Ra values indicate the mean of surface roughness before and after immersion, while ΔE* values refer to the average color changes, which were calculated by the aforementioned equation (Line 186-188).  In this study, we have examined the effect of khat on glazed and polished ceramic surfaces, but we did not compare the values of surface roughness before and after immersion. This can be considered as a limitation of this study and will considered in our future research studies. A clarifying statement has been added Line # 365-367 in the discussion section.

Comment:

 In Table 3, ‘Natural teeth group’ has two ΔE results of 2.12 (0.201) and 3.45 (.411).

Answer:

One is referring to (before immersion) and the second is referring to (after immersion) as pointed in Table-3.

Comment:

The ΔE results of 2.12 (0.201) is for control group? If it is for control group, ‘before immersion (Line 220)’ is not correct expression, because the control group will not be immersed.

Answer:

The expression “control group” is now removed from the manuscript.

Comment:

In addition, why is the ΔE value of the control group of natural teeth so high as 2.12?

Answer:

Were the study was conducted, the natural teeth have high level of fluorosis as seen in Figure-1 A. This is now mentioned in the manuscript with reference (line # 318-317).

Comment:

 The p value 0.000 should be expressed as <0.001

Answer:

We have changed it accordingly in the whole manuscript.

Comment:

 P within group’ is a strange expression. What group do you mean? You have many kinds of groups including subgroups. Readers will be confused.

Answer:

We have changed it accordingly in the both tables.

Comment:

 Line 126) All samples were glazed? Including polishing group? Glazed and then polished?

Answer:

Yes, all samples were glazed. Then to mimic the actual clinical statues, half of the samples were not touched, and other half of the samples of the subgroup two were polished. As mentioned in materials and methods.

Comment:

Line 133 and 140) Did the polishing group of zirconia also get glazing and then polished?

Answer:

Yes, it was glazed, then half of them were polished (as mentioned in line # 132 and 141)

Comment:

 You should revise not only Table 1 but also your entire manuscript regarding the Vitablocs Mark II and Vita Easyshade V.

Answer:

It has been revised as suggested in the whole manuscript.

Comment:

Line 96 and 247) I think the null hypothesis should be reversed and rejected based on the p-values you expressed in footnotes of Tables 2 and 3.

Answer:

We agree, therefore, it has been revised accordingly (Lines 96-98 and 258-260)

Comment:

 Line 214) ‘Significantly differed’ and ‘P>0.05’ ?

Foot notes of Table 2 and 3: Different superscript letters indicate statistically significant difference inside the respective subgroup (P < 0.05)

 Line 131) CAC/CAM --> CAD/CAM

Answer:

It has been changed accordingly

Comment:

Finally I suggest to refer and insert the following recent papers regarding the evaluation of color and roughness changes of restorative materials:

Answer:

The suggested references are now added and discussed in the manuscript (Line#328-337).

Round 3

Reviewer 3 Report

Thank you for revising the manuscript. I think it has improved a lot.

However, do you still think you have studied khat's effect on the surface roughness of feldspathic and zirconia porcelain materials? I don't think so. You have averaged the surface roughness values ​​before and after immersion, so you cannot know the effect of khat.

For example, if the material A has a surface roughness of 0 before immersion and then increases to 10 after immersion, the mean of surface roughness before and after immersion is 5. If the surface roughness of material B is 5 before immersion and then remains 5 without change after immersion, the mean is also 5. The surface roughness of material A was increased by 10 by immersion, and B was unchanged, but both averages were equal to 5. So, you cannot know the effect of immersion at all. 

If you wanted to know the effect of khat, you shouldn't have averaged the surface roughness before and after immersion, but you should have calculated the difference. (Or you could have created a control group and compared it, or calculated the statistical difference between before and after immersion.)

Thus, you have never studied the effect of khat on the surface roughness of the ceramic samples. The results in Table 2 and the results you presented in the abstract show the effect of polishing on the surface roughness of ceramics, or the difference in surface roughness among tested ceramic materials. The impact of khat is not recognizable.

This means that your manuscript's title, abstract, and entire manuscript must be revised. For example, in (Line 28-29) "Results revealed that the Ra and ΔE* values of the different types of tested porcelain were influenced by KH" and (Line 34-35) "KH significantly affected both surface parameter and color of glazed or polished porcelain materials and natural teeth", how do you know that Ra values were influenced by KH? How do you think about the title of your manuscript?

Author Response

RESPONSE:

We are very grateful for the valuable comment of the reviewer. We fully agree with his point, therefore, we have gone back to the raw data obtained through this study. The values of glazed and polished surfaces before and after khat immersion were obtained and drawn in Figure-5. The Figure shows the obvious differences of values before and after immersion in KH, all the results were significant differences with p value more 0.001.
A statement is now added in the results and in the discussion sections (highlighted in RED).